# Acellular Dermal Matrix Used in Diabetic Foot Ulcers: Clinical Outcomes Supported by Biochemical and Histological Analyses

**DOI:** 10.3390/ijms22137085

**Published:** 2021-06-30

**Authors:** Ferdinando Campitiello, Manfredi Mancone, Marcella Cammarota, Antonella D’Agostino, Giulia Ricci, Antonietta Stellavato, Angela Della Corte, Anna Virginia Adriana Pirozzi, Gianluca Scialla, Chiara Schiraldi, Silvestro Canonico

**Affiliations:** 1Department of Advanced Medical and Surgical Sciences, University of Campania Luigi Vanvitelli, Piazza Miraglia 2, 80138 Naples, Italy; ferdinando.campitiello@unicampania.it (F.C.); angeladellacorte@live.it (A.D.C.); Dr.scialla@gmail.com (G.S.); silvestro.canonico@unicampania.it (S.C.); 2Department of Experimental Medicine, Section of Biotechnology, Medical Histology and Molecular Biology, University of Campania Luigi Vanvitelli, Via L. De Crecchio 7, 80138 Naples, Italy; marcella.cammarota@unicampania.it (M.C.); antonella.dagostino@unicampania.it (A.D.); giulia.ricci@unicampania.it (G.R.); antonietta.stellavato@unicampania.it (A.S.); adripirozzi@gmail.com (A.V.A.P.); chiara.schiraldi@unicampania.it (C.S.)

**Keywords:** diabetic foot ulcer, biomaterials, biochemical and histological analyses

## Abstract

Diabetic foot ulcer (DFU) is a diabetes complication which greatly impacts the patient’s quality of life, often leading to amputation of the affected limb unless there is a timely and adequate management of the patient. DFUs have a high economic impact for the national health system. Data have indeed shown that DFUs are a major cause of hospitalization for patients with diabetes. Based on that, DFUs represent a very important challenge for the national health system. Especially in developed countries diabetic patients are increasing at a very high rate and as expected, also the incidence of DFUs is increasing due to longevity of diabetic patients in the western population. Herein, the surgical approach focused on the targeted use of the acellular dermal matrix has been integrated with biochemical and morphological/histological analyses to obtain evidence-based information on the mechanisms underlying tissue regeneration. In this research report, the clinical results indicated decreased postoperative wound infection levels and a short healing time, with a sound regeneration of tissues. Here we demonstrate that the key biomarkers of wound healing process are activated at gene expression level and also synthesis of collagen I, collagen III and elastin is prompted and modulated within the 28-day period of observation. These analyses were run on five patients treated with Integra^®^ sheet and five treated with the injectable matrix Integra^®^ Flowable, for cavitary lesions. In fact, clinical evaluation of improved healing was, for the first time, supported by biochemical and histological analyses. For these reasons, the present work opens a new scenario in DFUs treatment and follow-up, laying the foundation for a tailored protocol towards complete healing in severe pathological conditions.

## 1. Introduction

Tissue repair is a dynamic and interactive process that involves soluble mediators, extracellular matrix, blood vessels, blood cells and several other cell types of connective and epithelial origin. The physiological response to skin damage is also characterised by temporally successive and tightly interdependent steps, namely haemostasis, inflammation, proliferation and remodelling [1]. Failure in control or excessive duration of one of the previously mentioned phases can compromise wound healing, causing the onset of chronic wounds. Platelets play an important role in the beginning of the wound repair, being responsible, besides their haemostatic role, for the biosynthesis and release of growth factors and cytokines (PDGF, TGFβ-1, FGF, EGF, VEGF etc.), which are, in turn, key biochemical regulators of the healing cascade. These soluble mediators indeed attract leukocytes and endothelial cells to the wound. Notably, among them, transforming growth factor beta 1 (TGFβ-1) represents a recruiting signal that attracts neutrophils towards the lesion site, lowering the possibility of the onset of skin infections [2]. Besides its chemotactic activity, TGFβ-1 is also known to stimulate angiogenesis, extracellular matrix deposition and fibroblasts migration/proliferation [3]. The transient expression of TGFβ-1 is related with wound healing inflammatory phase together with TNF-α expression and bioavailability [4]. Inflammation is recognized as an important step of the wound healing process, preparatory to activation of the proliferative phase. Chronic and acute wounds are characterised by excessive inflammation, enhanced proteolysis and reduced matrix deposition. In fact, TGFβ-1 and TNF-α appear inversely regulated in chronic wounds being TGFβ-1 downregulated and TNF-α up-regulated [4]. Among the growth factors, VEGF is known to affect wound healing process not only in relation to angiogenesis, that is very important for novel tissue viability, oxygenation and functionality, but also for collagen deposition and matrix assembly. VEGF is expressed by diverse cell types such as platelets, neutrophils, macrophages and, in turn, it stimulates metalloproteinase (MMPs) expression in both endothelial and vascular smooth muscle cells [5,6]. MMPs play a role in balancing synthesis and degradation of ECM towards remodelling, allowing neutrophils invasion, promoting the angiogenesis, thus contributing to maintain physiological tissue homeostasis. In physiological condition, the inflammation mediators release and the inflammatory cells recruitment, is followed by fibroblasts proliferation. Then endothelial cells and new blood vessels, within extracellular matrix, are able to produce granulation tissue that fills damaged tissue. During the remodelling phase, the scar is softened without loss of resistance to restore physiological functions avoiding fibrosis. In diabetic disease, ulcers do not follow an orderly progression of wound healing due to many patho-physiological factors that negatively affecting the process. Diabetic foot ulcers (DFUs), despite the Standard of Care for treatment [7,8,9], need a longer timeframe to close, and often do not repair. They are featured by higher rates of infection and, in the worse but unfortunately not rare cases, lower limb amputation. Recently introduced dermal substitutes act as extracellular matrix components supporting the differentiation and proliferation of the cells involved in the healing process [10,11,12]. As a result, dermal substitutes in the treatment of DFU patients reduce healing times and infection rates [13]. These results in lower incidence of major amputations, allowing the surgeon to perform minor amputations in an attempt to preserve foot posture [13,14,15]. Many studies have been run on the use of Integra^®^ sheet on superficial wound (Integra^®^ Dermal Regeneration Template; IDRT); however, this dermal substitute has to be placed in rather plane lesions [16,17]. As a matter of fact, there is a very large number of deep lesions in DFUs, involving different tissues from epithelium to cartilage or bone structures. Integra^®^ Flowable Wound Matrix (IFWM) is a dermal substitute that finds its indication mainly in the treatment of deep wounds including surgical wounds and diabetic ulcers of both partial and full thickness varieties.

In the present paper, we report about biochemical, histological and finally clinical outcomes obtained by using IDRT, for plane lesions, and IFWM, for deep cavitary lesions, in DFU patients. The devices proposed in this study are based on a collagen and shark derived chondroitin sulphate matrix (chondroitin 6-sulfate as reported by the manufacturer). IDRT is a semi-biological implant consisting of two layers; the upper layer is a silicone sheet that acts to protect the deeper layer that consists of a protein matrix [18]. IFWM comes in a sub-maximal swelling condition, while a specific mixing device based on two syringes permit the Flowable matrix to be mixed with saline solution under sterile conditions before application on/in the lesion. The latter shows a certain plasticity, it can spread in the void volume of the lesion, to fill it, inducing tissue regeneration.

The goal of our work was to evaluate the performances of these two different class medical devices on the lesion’s treatments. Beside patient’s clinical follow-up, the research aimed at evaluating the healing progression at histological level and, at the same time, the expression of specific genes (TGFβ-1, TNF-α, MMPs, VEGF, Collagen I and Collagen III) involved in different phases of tissue repair. Herein we propose an “in process control” of the healing progress based on a “molecular follow-up”, beside the histological and clinical one, allowing a better understanding of the role of IFWM or IDRT application in the wound site, and shedding light on the molecular action mechanisms of these biomaterials within the regenerative process.

Particular interest was given to the reduction of inflammation, stimulation of fibroblasts, deposition of collagen, formation of vessels, production of new extracellular matrix and final tissue repair.

Clinical key points of the study were average time to complete DFU closure, healing rate evaluation at 16 weeks, hospitalization period, time frame between surgical procedure and walking.

## 2. Methods and Materials

### 2.1. Patients, Surgical Methods, and Clinical Outcomes

The present study was carried out in compliance with the principles laid down in the Declaration of Helsinki, in agreement with the international conference on harmonization good clinical practice guideline, and in accordance with applicable regulatory requirements. All patients with diabetes with neuropathic lesions of the foot observed at the General and Geriatric Surgery Unit of the University of Campania “Luigi Vanvitelli”, Naples, Italy between April 2017 and January 2019 were evaluated for inclusion in the present study. All enrolled patients signed written consent to receive treatment with the IDRT or IFWM after receiving adequate information about the study.

Inclusion criteria were male and female patients with Type I or II Diabetic >18 years of age, who had neuropathic lesions of the foot (Wagner grade 3), diagnosed at least 1 month before the start of the study, with a body mass index (BMI) ≤ 30 kg/m^2^ and ankle-brachial index (ABI) ≥ 0.8 and HbA_1c_ less than 12%. Exclusion criteria included a Michigan Diabetic Neuropathy Screening Score (MDNS) < 3 an ABI ≤ 0.8; coagulopathies and/or self-immune diseases. Neuropathy screening consisted of a 15-item questionnaire on and a brief clinical examination; patients with a positive score > 2 on clinical examination were considered to have neuropathy [19]. Clinical examination and ABI measurement were used to evaluate patients’ vascular assessment. Patients with an ABI ≤ 0.8 were considered to have peripheral vascular disease [20].

Ten patients with DFUs involving different tissues from epithelium to cartilage or bone structures (Wagner grade 3) were selected to receive acellular dermal matrix treatment. Five with flat lesions without cavities were covered with IDRT and the other half of the lesions, which presented with cavities, were treated with IFWM, a gel filling the cavity completely.

All patients were subjected to biopsy of the lesion to determine bacterial culture with antibiogram and Minimal Inhibitory Concentration (MIC). Targeted Antibiotic treatment started 7–10 days before surgery, depending on the type of microorganisms, and continued until the wound healed. Plain radiograph of the wound site was performed to assess the involvement of deep structures at enrolment. Clinical examination, laboratory blood tests, standardized photography and wound biopsy of wound were executed at t_0_ and interval times defined by the flowchart. All patients underwent a local infiltration of 20 cc mepivacaina cloridrato 1%.

In the IDRT groups (Figure 1), a Hydrosurgery System debridement of soft tissue and tendons on lesion was performed in 4 patients and a surgical ultrasonic debridement of soft tissue and tendons and bone cutting in 1 patient. The IDRT bilayer was applied in 5 patients with clips and covered with wet gauze, in T2 the silicone layer is removed, in T4 it was treated with reconstructive surgery with split-thickness-skin-graft (STSG) if no healing had occurred.

The patients of IFWM group (Figure 2) underwent an abscess drainage, wounds were first washed with normal sterile saline solution, and a spoon of Volkmann debridement of soft tissue and tendons.

The IFWM, consisting of cross-linked collagen and chondroitin 6-sulfate, is supplied in a kit containing a sterile dry collagen particle syringe, an empty saline syringe, a luer lock connector and a cannula directly into the cavity after surgical debridement. The two syringes are connected, and the collagen mixed with 3 mL of saline solution until a homogeneous consistency product is obtained. The product through an angiocath is injected filling the cavity lesion without leaving space. After application of IFWM, surgical wound edges were either approximated with stitches or left open to allow healing by secondary intention (Figure 2) and covered with wet gauze.

An inelastic multilayer multicomponent bandage was used as compression therapy. It was recommended to patients not to walk and upload for two weeks; subsequently, a limited ambulation was allowed using a removable offloading device.

The postoperative follow-up outpatient was scheduled with clinical examination, standardized photography, and inflammation marker (edema, erythema, increased local temperature, presence of abscesses) assessment.

Biopsy of wound, indicated as pathological control, have been executed for the patients selected at enrolment (T_0_), at T_1_ (7 days), T_2_ (14 days), T_3_ (21 days), T_4_ (28 days). In addition, for each patient simultaneously with the first biopsy (T_0_), a fragment of healthy skin, indicated as CTR, at a distance of 10 cm from the lesion was isolated.

The pathological control biopsies, not to compromise wound healing, were very small (1 mm); were picked by the surgeons in marginal area in IDRT group and while in the centre of the lesion in the IFWM group.

Laboratory blood tests (glycaemia, WBC, HbA1c, ESR, PCR) have been executed at enrolment at T_2_ (14 days), T_4_ (28 days). During every follow-up visit, the binary presence or absence of each inflammation marker (edema, erythema, increased local temperature, presence of abscesses) was recorded in T_1_–T_4_ (Table 1). Adverse events were recorded. The state of healing was assessed by clinical examination, and final healing was defined as 100% re-epithelialization of the wound.

### 2.2. Quantitative Real Time PCR (qRT-PCR) on Tissue Biopsies

Analyses were performed slightly modifying previously reported protocols [21]. Specifically, total RNA was extracted from biopsies using TRIzol^®^ (Invitrogen, Milan, Italy) and Tissue Ruptor homogenizer (Qiagen, Hilden, Germany). Retrotranscription was performed using Reverse Transcription System Kit (Promega, Milan, Italy) in order to analyse the expression levels of TGFβ-1, TNF-α, VEGF, MMP-2, MMP-9, MMP-13, COL-I and COL-III. qPCR was then performed using appropriate primer pairs reported in Table 2, and iQ™ SYBR—Green Supermix (Bio-Rad Laboratories Srl, Milan, Italy). Normalized gene expression was calculated as the fold-change exploiting the comparative threshold method (ΔΔ Ct = difference of Δ Ct between healthy control at time 0 and damaged or pathologic tissue at different time). The results were normalized to the housekeeping gene hypoxanthine guanine phosphoribosyltransferase (HPRT), and expressed as fold change expression, the output was calculated by the Bio-Rad iQ™5 software (Bio-Rad Laboratories Srl). Data represent distribution of 5 patients (each with respect to the expression of the specific marker in the healthy tissue of the patient before treatment) for each gene in the time course (up to 28 days) normalized (each one with respect to the expression of the specific marker in the healthy tissue of the patient before treatment). PCR experiment were performed in triplicate for all patients treated either with IDRT [5] or with IFWM [5].

### 2.3. Western Blotting

Protein extracts were obtained from biopsies using radioimmunoprecipitation lysis assay (RIPA buffer) (1×) (Cell Signaling Technology, Denver, CO, USA) and homogenizing the tissues by Tissue Ruptor homogenizer (Qiagen, Hilden, Germany). For western blot experiments, protein concentration was quantified by using Bio-Rad protein assay reagent (Bradford method, Bio-Rad Laboratories, Milan, Italy). Then equal amounts of protein (20 µg) were loaded on 10% SDS-PAGE (Sodium Dodecyl Sulphate—PolyAcrylamide Gel Electrophoresis), and then transferred to nitrocellulose membrane (GE Healthcare Life Sciences, Milan, Italy) as previously described [19]. Nitrocellulose blocking was obtained in 5% *w*/*v* powder skimmed milk (Microtech, Naples, Italy) dissolved in Tris-buffered saline with 0.05% Tween (TBST) for 1 h at 4 °C. COL-I (180 KDa), COL-III (120 KDa), Elastin (70 KDa) (48 KDa). All primary antibodies were used at 1:200 dilution and incubated at room temperature overnight. The Actin (1:1000 *v*/*v* dilution) was used as housekeeping protein to normalize the specific protein levels. All antibodies purchased from Santa Cruz Biotechnology, Santa Cruz, CA, USA); COL-I sc-59772, COL-III sc-271249, Elastin sc-58756, Actin sc-8432). Washing of nitrocellulose membranes was performed three times for 10 min in TBST at room temperature (RT). Immunoreactive bands were detected by chemiluminescence by using corresponding horseradish peroxidase-conjugated secondary antibody (1:10,000 dilutions) (Santacruz Biotechnology, Santa Cruz, CA, USA) for 2 h. ECL system (Chemicon-Millipore, Darmstadt, Germany) was used to reveal chemoluminescent signals for each protein according to the manufacturer’s instructions. The semi-quantitative analysis of protein levels was carried out by ChemiDoc-It TM 500 Imaging System (Cambridge, UK).

### 2.4. Sample Preparation for Histology

Tissue biopsies were fixed in 2.5% glutaraldehyde in 0.1 M cacodylate buffer (pH 7.4) overnight at 4 °C, post fixed in 1% OsO_4_ and Uranil Acetate Replacement (Electron Microscopy Science), de-hydrated in ethanol and embedded in epoxy resin. The epoxy resin embedded samples were sectioned through an ultra-microtome Reichert E (Reichert, Heidelberg, Germany). Semithin sections (1 µm), obtained from epoxy resin embedded samples, were stained with toluidine blue, observed and photographed at Nikon Eclipse Ci using bright field optical microscopy.

### 2.5. Statistical Analyses

Data were analysed using SPSS version 16 for Windows (SPSS Inc, Chicago, IL, USA). The continuous variables were expressed as mean ± standard deviation, and categorical variables were expressed as frequencies (%).

### 2.6. Endpoints

The primary endpoints of our work consist in evaluating the healing progress following the gene expression trend of selected biomarkers such as TGF-β1, TNF-α, MMPs, VEGF and through histological evaluation, also collagens were analysed using western blotting. Finally achieving for each patient, a biochemical and cellular fingerprint. Secondary endpoints included the total healing time (100% wound closure), rate wound healing at 16 weeks, the interval (days) between the date of the intervention and the deambulation, the duration (days) of hospital stay.

## 3. Results

### 3.1. Surgical Method and Clinical Evaluation

Baseline demographics and clinical characteristics are shown in Table 3. A total of five applications of the IRDT were performed on five patients (four males and one female) with a median age of 61.20 ± 5.26 years. The ulcer lesion was averagely present 15.00 ± 5.20 month and in 60% it was localized in the midfoot and 80% of patients showed osteomyelitis. There were no major amputations, but one out the five patients underwent minor amputations. Results of laboratory blood tests of IDRT and IFWM are shown in Table 4. The most frequently found bacteria were Staphylococcus aureus in 60%, *Enterococcus faecalis* in 40%, *Pseudomonas aeruginosa*, *Staphylococcus epidermidis*, *Proteus mirabilis*, *Acinetobacter baumannii* and *Corynebacterium striatum* in 20%. The rate of wound healing after 16 weeks was 100% and time to healing was 56.00 ± 5.95 days (Table 5). In two applications of IDRT (Figure 3), a further approach was necessary with reconstructive surgery (STSG), while the other three patients obtained complete healing in T_4_ (Table 3). The hospitalization lasted 14.60 ± 3.97 days and the interval between surgery and the deambulation was 92.00 ± 40.87 days.

A total of five applications of the IFWM were performed on five patients (three males and two females) with a median age of 55 ± 3.74 years. The lesion had been present for an average of 7.00 ± 9.72 months and it was mainly localized in the midfoot, in addition three out of five patients showed osteomyelitis. There were no major amputations, and one out of the five patients underwent minor amputations. *Staphylococcus aureus, Pseudomonas aeruginosa* and *Staphylococcus epidermidis* were among the most frequently found bacteria (40% of lesions). The rate of wound healing after 16 weeks was 100% and time to healing was 34.00 ± 8.9 days (Table 5). In four applications, the matrix was applied filling the wound and the edges were approximated by sutures; healing occurred by secondary intention in 20% of patients (Figure 4, Table 5). The hospitalization time was 12.80 ± 3.27 days and the interval between surgery and the deambulation was 58.00 ± 8.37 days.

The acellular dermal matrix was well tolerated since the first dressing with no relevant inflammation marks, reduced exudates, edema was not occurring, neither erythema. No increase in local temperature or abscesses was not recorded. Only one patient treated with IFWM, in T_1_, produced exudate (i.e., serous and blood); however, already in T_2_ this was reduced. In the treated cases, no major amputations were needed, and an incidence of minor amputations was observed in two out of ten; seven out of ten patients treated had osteomyelitis.

The laboratory blood tests, executed at enrolment at T_0_ and T_4_, are shown in Table 4. In all patients, a single application of dermal matrix was performed, and all patients recovered normal walking.

### 3.2. Gene Expression Analyses of Specific Tissue Recovery by Using qRT-PCR

Specific biomarkers were selected to evaluate the bio-reparation stages occurring in the lesions of the treated patients. In particular, to figure out the inflammatory status, that is known to be the preliminary phase in wound healing, TNF-α and TGFβ-1 gene expression was evaluated. Angiogenesis stimulation in the wound biopsies was studied by vascular-endothelial growth factor (VEGF) expression. Finally, to be aware of the extracellular matrix remodelling and deposition during healing, the gene expression of metalloproteinases 2, 9 and 13 were evaluated together with COL-I and COL-III. All the analyses were performed by qRT-PCR (Figure 5, Panel I and Panel II).

The effect of IDRT application on the expression of the biomarkers reported above was evaluated in the healing process of superficial lesions of five different DFU patients (Figure 5 Panel I). Specifically, we observed that TNF-α expression showed high distribution in the wound sample with respect to healthy control; however, on average, about a five-fold increase is found; the patients distribution seems narrowed during healing and the TNF-α expression was reduced, resembling, on average, the healthy control at time zero. Differently, TGF-β1 showed a trend of increase of the means of the distributions up to 21 days (about four-fold with respect to the healthy control) and it was successively down-regulated at 28 days. VEGF showed an up-regulation of six-fold at 14 days, then its expression was reduced, reaching an averaged value of 3.5-fold with respect to the healthy control at 21 days. Finally, after 28 days, the expression levels of VEGF turned similar to the control. MMP-2 expression and MMP-9 showed an upregulation of about four- to five-fold with respect to the healthy control, while they proved reduced at 28 days. On the contrary, MMP-13 presented high expression levels from the beginning, but, similarly to the other metalloproteases, its expression was reduced towards the healthy control level at 28 days. A narrower distribution among patients can be highlighted during the healing course. Finally, extracellular matrix proteins, such as collagens (type I and III), were also assessed. As shown in Figure 5, Panel 1, COL-I expression results higher than COL-III; its levels increase at 14–21 days about 5-fold, but it was slightly lowered at 28 days. Whereas COL-III showed the same level expression of healthy control up to 14 days and then it increased up 3-fold by 28 days.

The effect of the IFWM injection in the cavitary lesions on the inflammation and on the reparation/remodelling biomarkers was evaluated in the healing process of deep lesions of five different DFU patients (Figure 5 Panel II).

We observed that TNF-α expression is high at early times (T_0_ and seven days) but it is progressively reduced during the follow-up period. The TGFβ-1 mRNA increased at 14 days (approximately four-fold with respect to the healthy control) and with lower extent in the mean value between 21 and 28 days. VEGF expression presented a significant up-regulation of eight-fold at 21 days, whereas it appeared comparable with the healthy control values at the other time points. The expression of MMPs increased in the time course up to 21 days. Specifically, MMP-2 presented an onset before MMP-9 and MMP13, the latter is showing a minor upregulation and reduction only at the last time point (28 days). On average, MMP-2 was up-regulated about 18-fold vs. the healthy control in the first three weeks of observation. The MMP-9 expression was increased up to 14 days about seven-fold with respect to the healthy control; it remained high at 21 days (about five-fold) and it was reduced at 28 days to the control level. MMP-13 showed small fluctuations around the healthy control level for the whole time course investigated. Finally, collagens (type I and III) that are involved in ECM remodelling and assembly were also evaluated and reported in Figure 5 Panel II. Gene expression results for both collagens reported an increase up to 28 days, in particular COL-I of five-fold and COL-III of about two-fold.

### 3.3. Western Blotting

COL-I, COL-III and elastin protein expression were also evaluated through western blots. Figure 6 showed densitometric analyses of the patient chosen as the representative of those analysed for both groups of patients treated with IDRT (Panel I) or IFWM (Panel II). There is a common trend in increasing expression of selected markers, even if this increase is more evident in patients treated with IFWM. In detail, COL-I and COL-III expression reached the maximum at 14 days and then stabilized, maintaining a quite high expression level. Noteworthy, in IDRT-treated patients, at 28 days, COL-I stabilized to the same level of control. The latter is a consistent and relevant finding, since high accumulation of COL-I could result in the onset of a fibrotic scar. Elastin maintained high expression levels up to 28 days, consistently with the regenerative action observed towards a sound dermal tissue. Differently, in IDRT-treated patients, both COL-I and COL-III markedly increased their expression, while elastin expression was slightly up-regulated.

### 3.4. Histology and Ultrastructural Analyses

Histological analyses allowed us to observe the gradual healing process of IDRT- or IFWM-treated lesions (Figure 7). Notably, the histological healing features are slightly different among patients, and the histological recovery appears temporarily slower in IFWM patients with respect to IDRT ones. To show the variability observed among patients during the follow-up, in Figure 7 we reported, on the left-side, histological images of the worse healing degree that we observed for each time point, and on the right-side, histological images of the better performing repair for each time point.

In detail, Figure 7B (Panel I and Panel II) shows representative images of histological appearance of bioptic samples withdrawn before surgical treatment, where a strongly damaged connective tissue featured by red and white blood cells out from the blood vessels and degenerated extracellular matrix was present. The histological follow-up of IDRT- and IFWM-treated lesions let us observe a progressive reduction of inflammation, demonstrated by the gradual decrease of migrating leukocytes, together with the decrease of cellular debris, disrupted blood vessels, and damaged extracellular matrix (Figure 7, Panel I and Panel II (C–J)).

The histological analyses allowed also to observe, fibroblast activation (recognizable by the euchromatic nuclei) (Figure 7, Panel I (D,E); Panel II (F,G)), and newly synthesized matrix deposition. The activation of fibroblasts is temporarily earlier in IDRT-treated lesion with respect to the IFWM-treated ones. In the later phases of the follow-up, we showed endothelium barrier reconstitution, albeit, again, blood vessels are observable earlier in IDRT-treated lesions with respect to the IFWM-treated ones (Figure 7, Panel I (F,J); Panel II (J)). Notably, the presence of extravasating leukocytes was observed also at 14 days after IDRT treatment; in most cases, inflammation-activated endothelium was evident (Figure 7, Panel I (F)). Interestingly, fibroblast activation was less marked in the late phases of lesion follow-up, since these cells appeared to acquire gradually heterochromatic nuclei (Figure 7, Panel I and Panel II (H)) and, concomitantly, the extracellular matrix became compact due to de novo matrix biosynthesis, deposition/organization.

IDRT and IFWM were recognizable (yellow asterisks) within the histological analyses and appeared to act as recruiting substrates/scaffold at first for leukocytes (Figure 7, Panel II (C)), and later by connective cells towards colonization and tissue regeneration (Figure 7, Panel I (E,G,I)). In most cases, after 28 days IDRT and IFWM can be rarely observed (in agreement with a correct biodegradation of the matrices) and we noted a reconstituted healthy tissue (Figure 7, Panel I and Panel II (G,I)).

## 4. Discussion

Recently reported data show that diabetic patients are at high risk (25%) of developing a DFU [22]. DFUs have deleterious impact on the quality of life of diabetic patients, and, at the same time, on the health-care system, being that these morbidities are one of the main causes of hospitalization for diabetic patients. Notably, more than 50% of DFUs are known to become infected, increasing the risk or even need for surgery or amputation and thus increasing mortality incidence, with higher impact on patients and the health-care system [23]. Even the economic impact of DFUs is very relevant; in USA the incidence of cost for treating diabetic patients is doubled in the presence of DFU [24] and additional costs are represented by the reduced working capability of DFU patients. For these reasons, a more efficient approach with less invasive surgical methods is essential to reduce healing times, infection, amputation rates and post-operative recovery, allowing patients to restart an active life earlier. So far, the role of surgery in the treatment of deep wounds has been limited to cleansing infected and/or necrotic tissue to induce granulation and secondary healing tissue. However, a few years ago, Driver and co-workers [17] published a clinical trial exploiting IDRT in the management of DFUs, reporting an increased wound closure rate [17]. Numerous studies conducted on the use of IDRT have shown that the specific characteristics of the scaffold (pore structure, degradation rate and surface chemistry) could be fundamental in the formation of neodermis with functional elasticity similar to normal skin [14,25,26,27]. Some authors believe that Integra^®^ IDRT is the only acellular dermal matrix capable of inducing the formation of neodermis, considering other scaffolds instead “bioinductors” of the granulation tissue due to their short action as they degrade rapidly [28]. The presence of the chondroitin sulphate component increases the resistance of the scaffold against the action of collagenases [29] and allows to better control material half-life and improves cell–scaffold interaction. CS has been reported to have anti-inflammatory properties, showing a specific chemistry that leads to an optimal interaction between the scaffold and myofibroblasts responsible for the creation of new dermis [29,30]. Despite the clearly documented benefits, the clinical use of biobased scaffolds remains limited, thus it may be helpful to increase awareness of the advantages through systematic approaches that aim at correlating biochemical and biological features to clinical outcomes.

In our previous RCT [13], the use of biomaterials in the form of injectable gels for the treatment of DFUs showed a reduction in healing times and a less frequent major amputation and re-hospitalization. The closure of the lesion with sutures, after application of IFWM, allowed a reduced healing time, thus also avoiding generally occurring healing by “second intention”. The gel obtained after hydration of IFWM with saline solution allows, in the deep lesions, a more intimate contact of the matrix with the bed of the lesion, that can be filled up with this mouldable material with high plasticity. This product is easy to use, does not need donor sites and, to the best of our knowledge, is not responsible for undesired side effects, thus can be considered a new frontier in the DFU care.

In the light of these preliminary considerations, herein we report, for the first time, the comparative analyses of molecular, histological, and clinical observations of the healing process of DFUs after IDRT and IFWM applications. The study was conducted on 10 diabetic patients with foot lesions treated with IDRT (five patients) and IFWM injectable matrix (five patients), to evaluate the healing capability of these biomaterials and to unravel the mechanism underlying the reparation process.

In this report, the clinical results indicated high healing rate (100% patients) in a short healing time (IDRT, 56.00 ± 5.948 day; IFWM, 34.00 ± 8.94 day) and short hospitalization frames (IDRT, 14.60 ± 3.97 days; IFWM, 12.80 ± 3.27 days). In fact, the literature data report healing times between 12 and 20 weeks [31,32,33,34].

In the current experience, IDRT- and IFWM-treated lesions were rapidly colonized by fibroblasts, so that patients experienced fast wound closure. The absence of major amputations in the study and the low rate of minor amputations allowed the maintenance of plantar support. Minor amputations, when conservative treatment is hampered, are commonly considered an appropriate therapeutic target: In our experience, despite prolonged (and multiple) antibiotic treatments, it is not possible to completely eradicate osteomyelitis with non-surgical treatment. Shorter hospitalizations and the short interval between surgery and walking are among the positive outcomes for patients. The presented treatments could also lead to potential long-term savings, which should; however, be verified with a comprehensive cost analysis on a larger patients’ population.

The clinical outcomes are here supported by an extensive histological and biomolecular characterization.

We have verified that, within 28 days from the application of the material, inflammatory infiltrate is reduced. Endothelium barrier, disrupted at starting time, appears reconstituted. IDRT and IFWM are colonized by fibroblasts and, after about three weeks, give sufficient support to cell anchoring and proliferation. Extracellular matrix fibres appear more regularly deposited, progressively with the healing time. These histological observations suggest a good recovery of tissue turnover and revealed the ongoing tissue healing, as confirmed by the trend of TGFβ-1 and VEGF expression after IDRT and IFWM application. These markers are clear signals of tissue reconstitution, since they stimulate cell proliferation, differentiation, extracellular matrix production and vascularization [5,35].

Notably, the presence of blood vessels in tissue biopsies was found earlier in IDRT samples with respect to the IFWM ones, and this was in line with the early expression of VEGF expression in IDRT samples.

Increased TNF-α expression has been shown to improve chronic wound healing and diabetes-induced skin repair disorders in diabetic rats [4,36]. In a physiological pattern, an initial increase in TNF-α has to be followed by a slow decrease in its expression [4], that is the trend we found in the results herein reported (Figure 5).

The decrease of this inflammatory factor is supporting the action of the components of Integra^®^ matrices. In fact, it is worth noting that IDRT and IFWM contain chondroitin-sulphate (CS), whose anti-inflammatory properties are well known in different pathologies, such as osteoarthritis [37]. More recently, secretome analyses in an inflammatory model of osteoarthritis, revealed that CS is able to modulate inflammatory cytokines secretion [38]. CS was also found to improve wound healing in vitro [39].

However, it has to be considered that CS is only a minor component of Integra^®^ matrices (in terms of weight), entrapped in the cross-linked collagen fibres, nevertheless the 3D structure of the porous matrices and their geometry seem to be relevant for their beneficial role.

The consistency of TGFβ-1, VEGF and TNF-α expression trends during DFU patient’s healing time may propose these factors as effective molecular markers in the follow-up of DFUs.

However, further studies are needed also based on the analysis of circulating TGFβ-1, VEGF and TNF-α during the healing time of DFU patients to eventually confirm this hypothesis.

To study the tissue repair process, the synthesis and degradation of collagens in the matrix represent a key dynamic process that should be correctly balanced. In particular, collagen I and III are localized in fibrillar dermis and contribute to maintain structure, tissue integrity and tensile strength. In pathological conditions an unbalance of collagen synthesis may be found.

In fact, pathological phenomena, such as keloid formation or hypertrophic scars are accompanied by a persistent and unbalanced increase in the synthesis of these fibrillar proteins [40]. Studies of wounded skin and peripheral nerves in adult mammals treated with IDRT show that healing can be induced simply by appropriate control of wound contraction rather than scar formation [41]; clinical studies have shown improvements in scar quality in patients with hand burns treated with percutaneous IFWM with minimally invasive injection [42] and in burned breast reconstruction in patients with IDRT treatment [43].

Matrix metalloproteases (MMPs) play an essential role in wound healing as they regulate extracellular matrix degradation/deposition (remodelling) and cells’ migration in ECM. In chronic wounds, the deregulated expression of MMPs may inhibit wound closure preventing re-epithelialization. MMP-2 expression induces keratinocyte migration; MMP-13 degrades several collagens and promotes re-epithelialization indirectly by affecting wound contraction [6,43]. In our results, MMP-2, MMP-9 and MMP-13 showed an initial increase in their expression levels then decreased consistently with the concurring of tissue repair.

In this study, we report a slight fluctuation of collagens as quantified by western blotting; however, these values may indicate that a turnover of the fibrillar extracellular matrix occurs. The ongoing remodelling of tissues is positive to final healing.

In fact, at early stage, during the granulation tissue formation, COL-III is produced in large quantities by myofibroblasts, and this considerably unbalances the COL-I/COL-III ratio [42]. During the maturation of the wound, this ratio returns to physiological values. Taken together, qRT-PCR and western blot results support a balanced turnover of the fibrillar extracellular matrix toward a desired tissue remodelling and a successful healing.

Overall, the biochemical and biological analyses, performed on this cohort of patients affected by diabetic ulcers, indicated that Integra^®^ matrices application on DFUs drove a reduction of inflammatory markers and tissue inflammation and supported the physiological healing process, as demonstrated by the up regulation of specific molecular targets in a precise time-frame, avoiding the onset of chronic inflammation. The overall data reported are interesting and relevant for the clinical evaluation of the healing process, since it is known that the over-deposition of fibrillar matrix it generally driving towards fibrotic events that could compromise normal tissue repair [42].

The authors are aware of the limitations of this study, mainly related to the number of treatments/patients, and the inclusion in this cohort only of DFUs lesions that make impossible to compare the results obtained with other types of normal or chronic wounds. However, the limited number of patients enrolled permitted a multi-level characterization with diverse techniques, thus tackling the tissue regeneration issue with a wide and consistent number of biomarkers. Further studies, with a larger cohort and control groups, would help to determine the full implications of this new technique in the treatment of wounds.

## 5. Conclusions

Clinical evaluation of improved healing of DFUs was, for the first time, supported by biochemical and histological analyses during the regeneration time frame. Biopsy withdrawn at specific time intervals after surgery may help in assessing the healing stage and the quality of the new tissue. IDRT and IFWM provide effective tissue regeneration at molecular, histological and clinical levels, with a decrease in the inflammatory response. The results obtained represent a good starting point for exploring the acellular dermal matrix potentialities in tissue regeneration. Even if other studies are necessary to elucidate some aspects of the beneficial effect of these biomaterials in DFUs, the present data shed light on key biomarkers for molecular monitoring of the healing progress.

Timely repair is a critical point to reduce infection, hospitalization and amputation; all the latter may have a major impact on health care systems and thus affect social aspects, especially considering the aging of population in western countries.

## Figures and Tables

**Figure 1 ijms-22-07085-f001:**
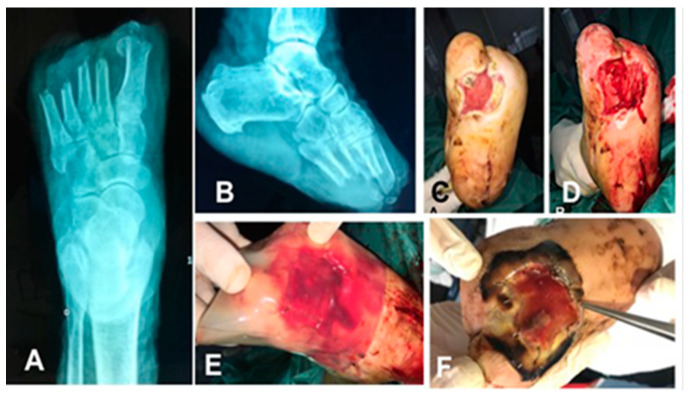
The plain radiograph of foot (**A**,**B**), a midfoot plantar injury on enrolment (**C**) and after debridement with hydro surgery system (**D**), treatment with IDRT (Integra^®^ Dermal Regeneration Template) fixed with metal clips (**E**), clinical examination at T1 (**F**).

**Figure 2 ijms-22-07085-f002:**
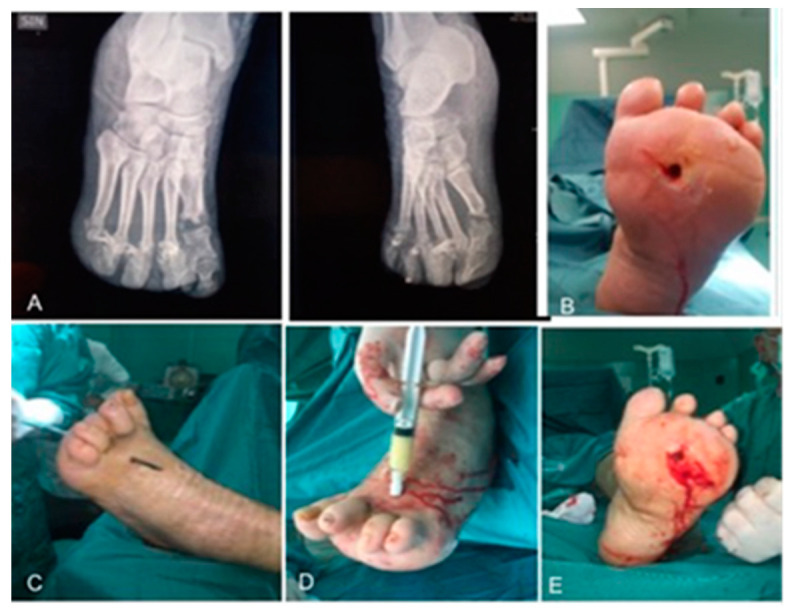
The plain radiograph of foot (**A**), a wound site at T0 with involvement of deep structures forefoot plantar lesion (**B**) and plantar-dorsal lesion (**C**). Description of the technique, IFWM (Integra^®^ Flowable Wound Matrix) is injected through a cannula (**D**,**E**).

**Figure 3 ijms-22-07085-f003:**
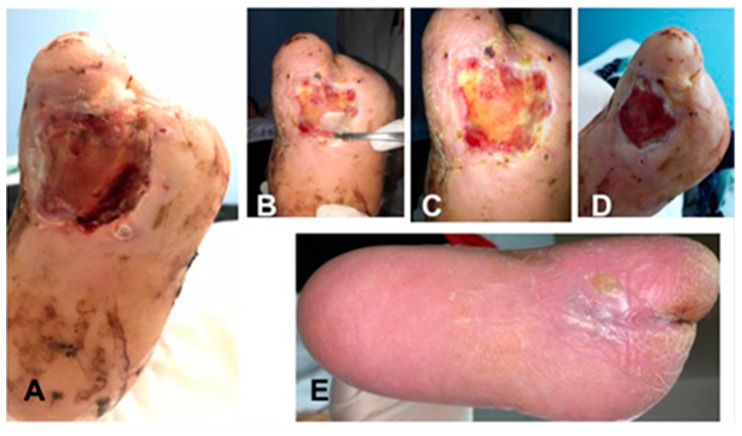
Lesion after treatment with IDRT at T_2_ before removal of silicone layer (**A**), at T_3_ after removal of silicone layer (**B**,**C**), at T_4_ (**D**,**E**), and the complete wound healing (**E**).

**Figure 4 ijms-22-07085-f004:**
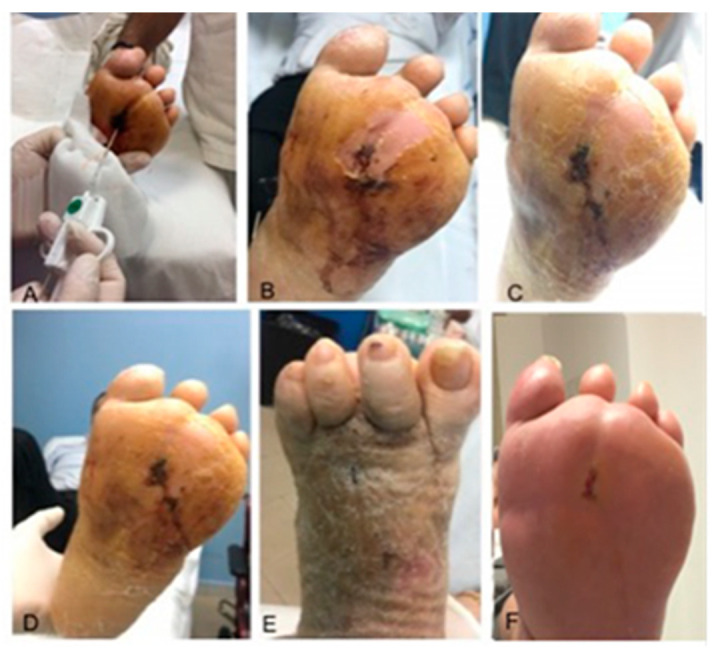
Lesion after treatment with IFWM at T_1_ (**A**), T_2_ (**B**), T_3_ (**C**), T_4_ (**D**,**E**), and the complete wound healing (**F**).

**Figure 5 ijms-22-07085-f005:**
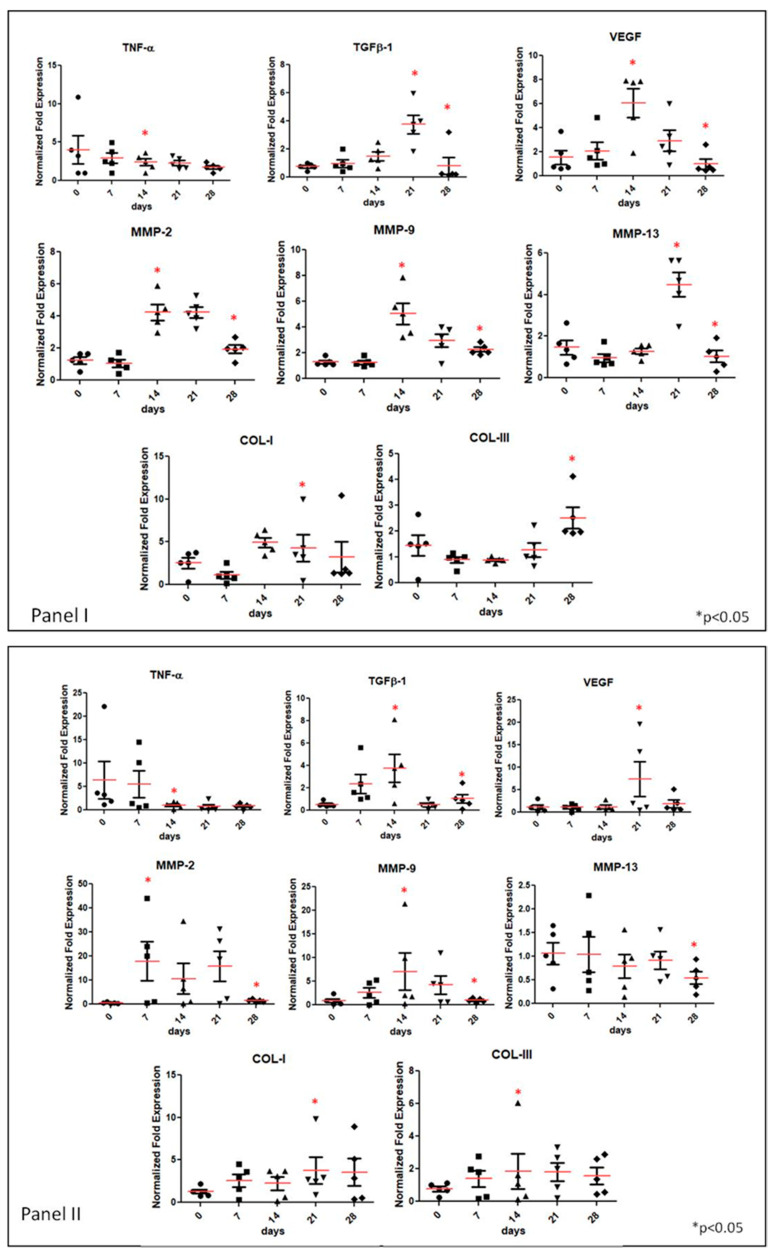
Gene expression profile provided by qRT-PCR on IDRT- and IFWM-treated lesions. Each of them is normalized with respect to its expression in the healthy tissue of the patient before treatment. (**Panel I**) Gene expression analyses carried-out by qRT-PCR, on TNF-α, TGFβ-1, VEGF, MMP 2–9 and 13, COL-I and COL-III in IDRT-treated lesions. Student *t*-test analyses were performed using graph-pad software and we compared two time groups as follows: TNF-α at 14 days vs. T0; TGFβ-1 at 21 days vs. T0 and 28 vs. 21 days; VEGF 14 days vs. T0 and 28 vs. 14 days; MMP-2 14 days vs. T0 and 28 vs. 14 days; MMP-9 14 days vs. T0 and 28 vs. 14 days; MMP-13 21 days vs. T0 and 28 vs. 21 days; COL-I 21 days vs. T0; COL-III 28 days vs. T0; statistical significance was indicated (*) for *p* < 0.05. (**Panel II**) qRT-PCR on biopsies of patients treated with IFWM. Student *t*-test analyses were performed using graph-pad software and we compared two time groups as follows: TNF-α at 14 days vs. T0; TGFβ-1, at 14 days vs. T0 and 28 vs. 14 days; VEGF 21 days vs. T0; MMP-2 seven days vs. T0 and 28 vs. seven days; MMP-9 14 days vs. T0 and 28 vs. 14 days; MMP-13 28 days vs. T0; COL-I 21 days vs. T0; COL-III 14 days vs. T0; statistical significance was indicated (*) for *p* < 0.05. Each biomarker is normalized as detailed in materials and methods section. Time zero refers to pathological control (normalized to healthy sample withdrawn from a region distant from the wound). Data represent distribution of five patients. qRT-PCR experiment was performed in triplicate.

**Figure 6 ijms-22-07085-f006:**
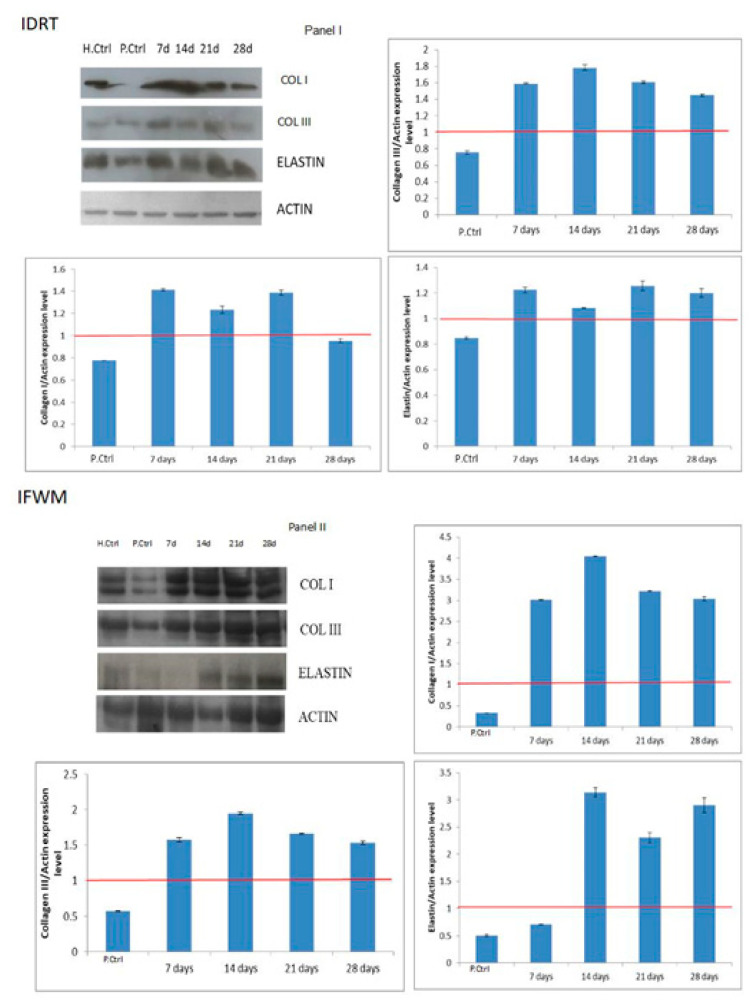
Western blot analyses of COL-I and III, and of elastin on IDRT- and IFWM-treated lesions during the post-surgical follow-up and relative controls. (**Panel I**) Western blot analyses on IDRT-treated lesions at different post-surgical time points, and on relative controls. H.Ctrl (Healthy Control) represents the sample withdrawn in the healthy part of the skin. P.Ctrl (Pathological Control) represents the sample withdrawn in the lesion before IDRT application. (**Panel II**) Western blot analyses on IFWM-treated lesions at different post-surgical time points, and on relative controls. H.Ctrl (Healthy Control) represents the sample withdrawn in the healthy part of the skin. P.Ctrl (Pathological Control) represents the sample withdrawn in the lesion before IFWM injection.

**Figure 7 ijms-22-07085-f007:**
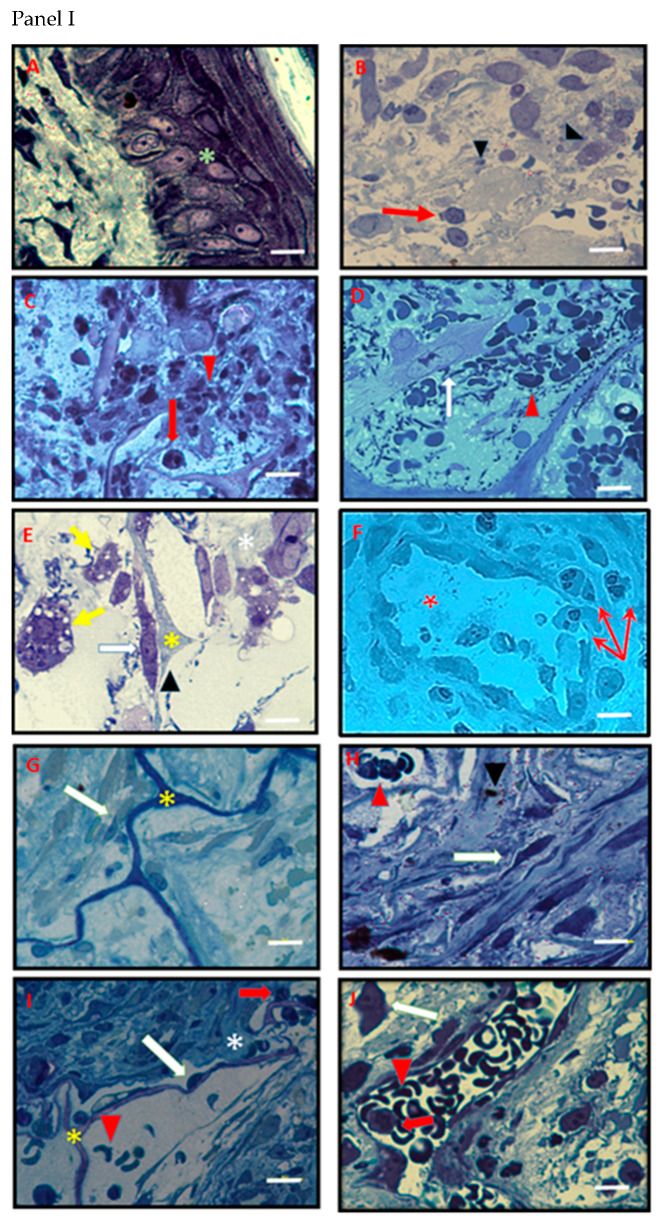
Histological features of tissue biopsies withdrawn at time point 0 and at different post-surgical time points after IDRT or IFWM application. **Panel I**: (**A**,**B**) Representative histological images of tissue biopsies withdrawn at time point 0 in a healthy part of the skin (**A**) or in the lesion (**B**); (**C**–**J**) representative images of the histological appearance of tissue biopsies withdrawn: Seven (**C**,**D**), 14 (**E**,**F**), 21 (**G**,**H**) and 28 (**I**,**J**) days after IDRT application. **Panel II**: (**A**,**B**) Representative histological images of tissue biopsies withdrawn at time point 0 in a healthy part of the skin (**A**) or in the lesion (**B**); (**C**–**J**) representative images of the histological appearance of tissue biopsies withdrawn: Seven (**C**,**D**), 14 (**E**,**F**), 21 (**G**,**H**) and 28 (**I**,**J**) days after IFWM application. The left images represent the lower degree of wound healing that we observed for each time point, whereas the right images represent the higher degree of wound healing that we observed for each time points. Green asterisk indicates epidermis, white arrows indicate fibroblasts; red arrows indicate leukocytes; yellow arrow indicates vacuolated somatic cells; red arrowhead indicates erythrocyte; black arrowhead indicates cellular debris; yellow asterisk indicates IDRT, white asterisks indicate extracellular matrix fibres, red asterisks indicate blood vessels. Scale bar = 10 µm.

**Table 1 ijms-22-07085-t001:** Flow chart.

Procedures	T_0_ (Enrollment)	T_1_ (1 Week)	T_2_ (2 Weeks)	T_3_ (3 Weeks)	T_4_ (4 Weeks)
Informed consent	X				
Clinical Examination	X (before debridement)	X	X	X	X
Laboratory Blood Tests	X (before debridement)		X		X
Wound Biopsy	X (before debridement)	X	X	X	X
Complications		X	X	X	X

**Table 2 ijms-22-07085-t002:** Primers sequence used for qRT-PCR analysis.

Gene Name (Symbol)	PCR Primer Sequence 5′ → 3′	Amplicon Length (bp)
Ipoxantina-guanina fosforibosiltransferasi (HPRT)	CATCCTGCACCACCAACTG CACAGTCTTCTGAGTGGCAG	117
Transforming growth factor, beta 1 (TGF-â1)	CTCGCAGCTGGAGGACTCC CTCGTCCAGGATGGCGTAG	103
Tumor necrosis factor alpha (TNF-á)	CGAGTGACAAGCCTGTAGC GGTGTGGGTGAGGAGCACAT	102
Matrix metallopeptidase 2 (MMP-2)	GCCGCCTTTAACTGGAGCAA TTCCAGGCATCTGCGATGAG	106
Matrix metallopeptidase 9 (MMP-9)	GCGCCACCACAGCCAACTATG TggATGCCgTCTATgTCgTCTTTA	104
Matrix metallopeptidase 13 (MMP-13)	TCCCTGAAGGGAAGGAGC CTCGTCCAGGATGGCGTAG	105
Vascular endothelial growth factor (VEGF)	AAGCTTGTGCGATGTTACCC CCAGGAGAGAATGTGGCAGT	110
Type I collagen (COL-1A1)	CAGCCGCTTCACCTACAGC TTTTGTATTCAATCACTGTCTTGCC	132
Type III collagen (COL-3A1)	TGGTCCCCAAGGTGTCAAAG GGGGGTCCTGGGTTACCATTA	125

**Table 3 ijms-22-07085-t003:** Baseline demographics and clinical characteristics. IDRT: Integra dermal regeneration template; IFWM: Integra Flowable wound matrix; STSG: Split-thickness skin graft.

	IDRT	IFWM
**Age (Years)**	61.20 ± 5. 26	55.00 ± 3.74
**Gender (Male/Female)**		
Females Males	1 (20%) 4 (80%)	2 (40%) 3 (60%)
**Body Mass Index (kg/m^2^)**	27.75 ± 3.70	29.06 ± 6.21
**Ankle Brachial Index**	0.94 ± 0.05	1 ± 0.12
Osteomyelitis Charcot Foot	4 (80%) 3 (60%)	3 (60%) 3 (60%)
Depth (cm) Length (cm) Width (cm)	------ 11. 40 ± 3.91 6.40 ± 2.41	8.40 ± 2.30 2.80 ± 1.92 2.00 ± 0.71
**Ulcer Location**		
Forefoot Forefoot-Midfoot	2 (40%) 3 (60%)	4 (80%) 1 (20%)
**Mean Duration of Ulcer (Months)**	15.00 ± 5.20	7.00 ± 9.72
**Surgical Debridement** **Surgical Debridement + Minor Amputation** **STSG**	3 (60%) 1 (20%) 2 (40%)	4 (80%) 1 (20%) ------

**Table 4 ijms-22-07085-t004:** Results of laboratory blood tests of IDRT and IFWM.

		T_0_	T_2_	T_4_
**IDRT**	**Glycaemia (mg/dL)**	185.80 ± 100.47	137.20 ± 28.98	132.40 ± 21.98
	**WBC (U/µL)**	8582.00 ± 1975.02	7700.00 ± 1396.42	7286.00 ± 1163.61
	**HbA1c (%)**	7.38 ± 1.12	6.74 ± 0.88	6.38± 0.99
	**ESR (mm/h)**	49.20 ± 23.57	36.00 ± 16.75	35.40 ± 22.58
	**PCR (mg/L)**	15.94 ± 24.79	6.70 ± 7.71	3.29 ± 3.83
**IFWM**	**Glycaemia (mg/dL)**	161.00 ± 31.10	133.80 ± 31.10	134.80 ± 21.32
	**WBC (U/µL)**	9298.40 ± 1989.11	8558.20 ± 1547,96	7996.00 ± 1547.96
	**HbA1c (%)**	7.86 ± 1.74	7.78 ± 1.73	7.26± 1.57
	**ESR (mm/h)**	52.80 ± 27.95	46.20 ± 24.09	41.40 ± 23.60
	**PCR (mg/L)**	7.54 ± 5.15	4.98 ± 3.99	2.97 ± 2.86

**Table 5 ijms-22-07085-t005:** Secondary endpoints in IDRT and IFWM.

	IDRT	IFWM
Time to Healing (Days)	56.00 ± 5.948	34.00 ± 8.94
Rate Wound Healing at 16 Weeks	100%	100%
Interval between Surgery and the Deambulation (Days)	92.00 ± 40.87	58.00 ± 8.37
Duration of Hospital Stay (Days)	14.60 ± 3.97	12.80 ± 3.27

## Data Availability

Data are present in figure and Appendix A further, data are available from the author upon request.

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
