# Peer review of "Acellular Dermal Matrix Used in Diabetic Foot Ulcers: Clinical Outcomes Supported by Biochemical and Histological Analyses"

_ijms, 2021, doi:10.3390/ijms22137085_

Round 1

Reviewer 1 Report

Novel applications of biologically active biomaterials provide exciting opportunities to improve the treatment of important clinical challenges. Along this direction, this paper describes an interesting clinical study on the application of acellular dermal matrices in DFU patients. Detailed information on how such of acellular dermal matrices regulate wound healing can help understand how and if they work and, as suggested by authors, guide their clinical application.

Major comments:

  • findings about IDRT, IFWM (e.g. table 5) should be compared against the current clinical practice. i.e. how does one know that 56+/-6 days time to heal is a substantial improvement or simply an efficient treatment. since you did not include a "control patient "group, you can seek data from the literature
  • I do not think it is correct to normalize the data of figs 5, 6 using results from the pathological control since I expect significant variation in the pathological control of different patients. I think results should be normalized via the healthy control dataset, which (in theory) should be more consistent among different patients.
  • Along the direction of previous comment, fig 5 should also contain the normalized pathological control result. In essence you compare how grafting alters what is happening in the pathological control sample (i.e. before treatment)
  • need to add statistical analysis on fig5 and fig 6 data (1 or 2 way anova + post-hoc test)
  • errorbars in fig 6 seem very small. Usually western blots have significantly larger errorbars. Maybe a possibility of some methodology error?
  • Due to the small number of samples, results in fig 5 and 6 should be plotted as box plots that include measurements as dots, not just as mean +/- sd
  • comment more about the relative sizes of lesions treated with DRT vs IFWM. in table 3 the depth of lesions in IFWM was indeed 8.4 cm? this sounds huge. Maybe you mean 8.4mm. also, you should provide the depth of DRT-treated wounds
  • collagen amount by itself is not a good marker of ECM remodeling. Maybe you can elaborate more on the structure of newly-synthesized collagen (dermis collagen differs from scar collagen)

Minor Comments

  • Authors need to improve text quality. There are several typos in the text (ie ERDT in 263). Authors can remove repeated text, for instance in the abstract
  • Check consistency of symbols. E.g. t0 (line 144) VS T0 (Table 1)
  • Improve caption text in figure 1,2. It does not make good sense
  • Provide more information about biopsy sampling. It should be clear that there is a “healthy control” and a ”pathological control”. I cannot find where the text describes the sampling of ”pathological control”
  • What are the conclusions of table 4 data?
  • I propose to organize the data of figure 5 according to gene (i.e. have a sub-figure that shows day 7, 14, 21, 28 data for both IDRT and IFWM). this can clarify results and also enable to compare IDRT and IFWM.
  • Similarly, I think fig 6 should be organize by protein.
  • Add the 1 baseline in all fig 5 subfigures
  • Presentation of results in 315-354 is confusing. You should either organize results by gene or by treatment
  • Add low-mag images in fig 7 to show the big picture. Low-mag pics show the “big picture” (i.e. what you mean by lower degree, higher degree)
  • The claim in lines 513-515 does not agree with fig 5. VEGF peaks in week 2 in IFWM and week 3 in DRT

Author Response

  • findings about IDRT, IFWM (e.g. table 5) should be compared against the current clinical practice. i.e. how does one know that 56+/-6 days time to heal is a substantial improvement or simply an efficient treatment. since you did not include a "control patient "group, you can seek data from the literature\

 As, for ethical reasons,  we could not not create a group of "control patients", according to the referee suggestions  we will refer, to the literature data that report healing times between 12 and 20 weeks.Following are reported additional references inserted in the manuscript

Katz IA, Harfan A, Miranda-Palma B, Prieto-Sanchez L, Armstrong DG, Bowker JH, Mizel MS, Boulton AJ: A randomized trial of two irremovable off-loading devices in the management of plantar neuropathic diabetic foot ulcers. Diabetes Care 28:555–559, 2005

Veves A, Falanga V, Armstrong DG, Sabolinski ML: Graftskin, a human skin equivalent, is effective in the management of noninfected neuropathic foot ulcers: a prospective randomized multicenter clinical trial. Diabetes Care 24:290–295, 2001

Margolis DJ, Allen-Taylor L, Hoffstad O, Berlin JA: Diabetic neuropathic foot ulcers: predicting which ones will not heal. Am J Med 115:627–631, 2003

Vas P, Rayman G, Dhatariya K, et al. Effectiveness of interventions to enhance healing of chronic foot ulcers in diabetes: a systematic review. Diabetes Metab Res Rev. 2020;36(S1):e3284. https://doi.org/10.1002/dmrr.3284

  • I do not think it is correct to normalize the data of figs 5, 6 using results from the pathological control since I expect significant variation in the pathological control of different patients. I think results should be normalized via the healthy control dataset, which (in theory) should be more consistent among different patients.

Thank you for your suggestion.

We have now normalized real time PCR data with respect to healthy control values at time zero and we have inserted the T0 (the pathological control) for the different patients for all genes investigated on the graphs. All the graph were modified accordingly and presented in a different format as also suggested by the referees

  • Along the direction of previous comment, fig 5 should also contain the normalized pathological control result. In essence you compare how grafting alters what is happening in the pathological control sample (i.e. before treatment)

As explained in the previous point, we have normalized our qRT-PCR results respect to healthy control data, for each of the gene analysed) and then we inserted pathological data on the graphs as T0.

  • need to add statistical analysis on fig5 and fig 6 data (1 or 2 way anova + post-hoc test)

Student T-test statistical analyses was performed on the basis of the most evident average variations to evaluate significance, for a specific marker in a specific time frame (i.e 0 vs 7days; 7 vs 21/28 days), in consideration on the overall modulation observed.

.

  • errorbars in fig 6 seem very small. Usually western blots have significantly larger errorbars. Maybe a possibility of some methodology error?

The data, reported as histograms, are referred to three densitometric measurements of the blots. In fact, the amount of protein extracted from very small biopsies only permitted to run one or 2 gels to be blotted

  • Due to the small number of samples, results in fig 5 and 6 should be plotted as box plots that include measurements as dots, not just as mean +/- sd

For figure 5, we have plotted new normalized values (vs healthy control at time zero, before treatments) on graph pad software in order to obtain a distribution of 5 patients (each with respect to the expression of the specific marker in the healthy tissue of the patient before treatment) for the time course (T0-28days). This normalization is additional to the internal one on HPRT, in agreement with the correct approach for a ΔΔct method.

  • comment more about the relative sizes of lesions treated with DRT vs IFWM. in table 3 the depth of lesions in IFWM was indeed 8.4 cm? this sounds huge. Maybe you mean 8.4mm. also, you should provide the depth of DRT-treated wounds

In fact, the cavity leasions are quite big, the purpose of the study is not a comparison between biomaterials, IDRT is a sheet biomaterial that we used to cover flat non-cavity (Wagner grade 3)  lesions and therefore the depth of the lesion was not described. IFWN is an injectable  gel (matrix)  that we have used in cavity lesions, in abscesses. these are lesions which unfortunately must be measured in centimeters and not in millimeters 

  • collagen amount by itself is not a good marker of ECM remodeling. Maybe you can elaborate more on the structure of newly-synthesized collagen (dermis collagen differs from scar collagen).

We agree with the referee that it would be interesting to have a follow up also on glyosaminoglycans, or fibronectin/elastin. However, in our experimental set up we have a multianalytical strategy implemented. The tissue quality was evaluated through molecular analysis on specific biomarkers (7 among cytokines, degradative enzymes, and fibrillar protein of the extracellular matrix) and on histological data, beside the protein quantification using western blotting. On the latter it has to be considered, that during healing for the selected diabetic ulcers the biopsy should be very small, thus to have a sufficient amount of proteins for the assay, we preferred to focus on collagen that anyway is the major constituent of the fibrillar part of the extracellular matrix.

Minor Comments

  • Authors need to improve text quality. There are several typos in the text (ie ERDT in 263). Authors can remove repeated text, for instance in the abstract

Thank you for your suggestion

I apologize but it is a typo this is IDRT

The abstract was modified according to the referee comments

 Abstract: Diabetic foot ulcer (DFU) is a diabetes complication which greatly impacts the patient’s quality of life, often leading to amputation of the affected limb unless there is a timely and adequate management of the patient. DFUs have a high economic impact for the national health system. Data have indeed shown that DFUs are a major cause of hospitalization for patients with diabetes. Based on that, DFUs represent a very important challenge for the national health system. Especially in developed countries diabetic patients are increasing at a very high rate, and, as expected, also the incidence of DFUs is increasing due to longevity of diabetic patients in the western population. Herein, the surgical approach focused on the targeted use of acellular dermal matrix has been integrated with biochemical and morphological/histological analyses to obtain evidence based information on the mechanisms underlying tissue regeneration. In this research report, the clinical results indicated a decreased postoperative wound infection levels and a short healing time, with a sound regeneration of tissues. Here we demonstrate that the key biomarkers of wound healing process are activated at gene expression level and also synthesis of collagen I, collagen III and elastin is prompted and modulated within the 28 days period of observation. These analyses were run on 5 patients treated with Integra® sheet and 5 treated with the injectable matrix Integra® Flowable, for cavitary lesions.  In fact, clinical evaluation of improved healing was for the first time supported by biochemical and histological analyses. This approach gave us the opportunity to establish the correlation among biochemical and cellular events, during DFU healing, and identify potential biomarkers for DFU follow-up. For these reasons, the present work opens new scenario in DFUs treatment and follow-up laying the foundation for a tailored protocol towards complete healing in severe pathological conditions.

  • Check consistency of symbols. E.g. t0 (line 144) VS T0 (Table 1)

I apologize but it is a typo this is T0

Improve caption text in figure 1,2. It does not make good sense

Figure1: The plain radiograph of foot (A-B), a midfoot plantar injury on enrollment (C) and after debridement with hydrosurgery system (D), treatment with IRDT fixed with metal clips (E), clinical examination at T1 (F).

Figure 2: The plain radiograph of foot (A), a wound site at T0 with involvement of deep structures forefoot plantar lesion (B) and plantar-dorsal lesion (C). Description of the technique, IFWM is injected through a cannula (D - E).

  • Provide more information about biopsy sampling. It should be clear that there is a “healthy control” and a”pathological control”. I cannot find where the text describes the sampling of ”pathological control”

In 176

The pathological control biopsies, not to compromise wound healing, were very small (1mm); were picked by the surgeons in marginal area in IDRT group and while in the center of the lesion in the IFWM group

What are the conclusions of table 4 data?

Statistical Analyses

Data were analyzed using SPSS version 16 for Windows (SPSS Inc, Chicago, IL). The continuous variables were expressed as mean ± standard deviation, and categorical variables were expressed as frequencies (%).

The mean difference of qualitative variables at t0, t2 and t4 were performed by ANOVA test. A p value of <0.05 was considered statistically significant.

There are no statistically significant differences between the laboratory blood tests, executed at enrollment at t2 and t4 with the exception of ESR with a decrease from time t0 at t4 (IFWM), however the number of patients was considered too small for any statistical indication

  • I propose to organize the data of figure 5 according to gene (i.e. have a sub-figure that shows day 7, 14, 21, 28 data for both IDRT and IFWM). this can clarify results and also enable to compare IDRT and IFWM.

The figures have been rearranged following the reviewrs suggestions.

However, please note that the aim of the study was not to compare IDRT and IFWM. In fact, the 1st can be used on superficial flat lesions, while the second preferably in void volumes/cavities that unfortunately may occur in the diabetic foot ulcers. Two panels in the same figure are therefore presented figures set will be presented for the 2 diverse devices/approaches and patients.

  • Similarly, I think fig 6 should be organize by protein.

We apologize, but don’t quite understand the suggestion of the referee, since there are western blotting referred to the same internal control (actin) thus presented as normally accepted by literature.

Then densitometries are reported to facilitate the readers idea of the modulation of the marker during the whole healing process from T0 to 28 days.

  • Add the 1 baseline in all fig 5 subfigures

Graphs have been changed and thus baseline may not be of interest anymore.

  • Presentation of results in 315-354 is confusing. You should either organize results by gene or by treatment

All results section was verified and then organized describing for each treatment the gene expression analyses correlated.

  • Add low-mag images in fig 7 to show the big picture. Low-mag pics show the “big picture” (i.e. what you mean by lower degree, higher degree)
  • As requested we added new panels with low magnification microphotographs of tissue biopsies, depicting the "big picture" of the sample histology. We insert these panels in the manuscript as "Supplementary material" to avoid redundant information in the main text. Even in these panels we insert images of the lower and higher degree of healing, to better describe the histological variability among patients.
  • The claim in lines 513-515 does not agree with fig 5. VEGF peaks in week 2 in IFWM and week 3 in DRT

VEGF results have been verified and corrected the text accordingly.

Supplementary material of histological analyses: low magnification histology of tissue biopsies withdrawn at Time 0 and at different post-surgical time points after IDRT or IFWM application.

In Panel I reports low magnification of histological analyses of patients treated with IDRT. In A and B representative histological images of tissue biopsies withdrawn at time 0 in a healthy part of the skin (A) or in the lesion (B) are reported. From C to L representative microphotographs of the histological appearance of tissue biopsies withdrawn at 7 (C,D), 14 (E,F), 21 (G,H) and 28 (I,L) days after IDRT application are shown.

In Panel II reports histological analyses of patients treated with IFWM. In A and B representative histological images of tissue biopsies withdrawn at time 0 in a healthy part of the skin (A) or in the lesion (B) are reported. From C to L representative microphotographs of the histological appearance of tissue biopsies withdrawn at 7 (C,D), 14 (E,F), 21 (G,H) and 28 (I,L) days after IFWM application are shown.

In both panels the left images (C,E,G,I) represent the lower degree of wound healing that we observed for each time point, whereas the right images (D,F,H,L) represent the higher degree of wound healing that we observed for each time points. Green asterisk indicates epidermis, white arrows indicate fibroblasts; red arrows indicate leukocytes; red arrowhead indicates erythrocyte; yellow asterisk indicates IFWM or IDRT, red asterisks indicate blood vessels.

Scale bar = 100 µm

Reviewer 2 Report

Notes for Authors:  In order to achieve maximum impact of their publication they  need to polish some issues found.

Introdution

  • Lines 59-64: You refer to the role of MMPs in the healing of ulcers. I suggest to reference recent meta analysis. (Tardáguila-García A, García-Morales E, García-Alamino JM, Álvaro-Afonso FJ, Molines-Barroso RJ, Lázaro-Martínez JL. Metalloproteinases in chronic and acute wounds: A systematic review and meta-analysis. Wound Repair Regen. 2019 Jul;27(4):415-420. doi: 10.1111/wrr.12717. Epub 2019 Mar 27. PMID: 30873727.)
  • Lines 109-116. In the introduction section I suggest ending it with the objective of the article, not with the support of its results.

Methods and Materials

  • Line 131: Refer to TASC II data for the diagnosis of peripheral vascular disease. (Norgren L, Hiatt WR, Dormandy JA, Nehler MR, Harris KA, Fowkes FG; TASC II Working Group. Inter-Society Consensus for the Management of Peripheral Arterial Disease (TASC II). J Vasc Surg. 2007 Jan;45 Suppl S:S5-67. doi: 10.1016/j.jvs.2006.12.037. PMID: 17223489)
  • Lines 130-131. Why are patients with mild neuropathy excluded according to the Michigan rating? (<3)
  • Line 138. Define the clinical type of ulcer to choose IFWF or IDRT
  • Line 140-141. Justify why the atb times were so long (continued until the wound healed)

Results

Table 3. Baseline demographics and clinical characteristics. Should include p values to show homogeneity or heterogeneity between groups

Table 5. Secondary endpoints in IDRT and IFWM. Should include p values to show homogeneity or heterogeneity between groups

Discussion

It is probably too long.

Lines 5589-556. I recommend again referring to the meta-analysis of the role of MMOPS in ulcer healing.

Author Response

 Introdution

- Lines 59-64: You refer to the role of MMPs in the healing of ulcers. I suggest to reference recent meta analysis. (Tardáguila-García A, García-Morales E, García-Alamino JM, Álvaro-Afonso FJ, Molines-Barroso RJ, Lázaro-Martínez JL. Metalloproteinases in chronic and acute wounds: A systematic review and meta-analysis. Wound Repair Regen. 2019 Jul;27(4):415-420. doi: 10.1111/wrr.12717. Epub 2019 Mar 27. PMID: 30873727.)

Thank you for your suggestion.

we have considered the referee's comments and will amend the text accordingly

 - Lines 109-116. In the introduction section I suggest ending it with the objective of the article, not with the support of its results.

Thank you for your suggestion.

The introduction was modified according to the referee comments

The goal of our work was to evaluate the performances of these 2 different class medical devices for treatments on the two diverse type of lesion. Beside patient’s clinical follow-up, the reasearch also focused on histological and biochemical analyses of these biomaterials effect within the regenerative process. Particular interest was given to the reduction of inflammation, stimulation of fibroblasts, deposition of collagen, formation of vessels, production of new extracellular matrix and final tissue repair. Clinical key points of the study were average time to complete DFU closure, healing rate evaluation at 16 weeks, hospitalization period, time frame between surgical procedure and walking. 

Our results support the use of Integra® dermal substitutes for DFU patients and shed new light on the steps that feature Integra-mediated wound healing, at clinical, histological, and molecular levels. Some differences exist in the molecular and histological aspects of wound healing process mediated by IDRT and IFWM, but both proved useful in sus- taining the lesion repair obtaining improvement in the wound treatments. Specifically, we demonstrate that both IDRT and IFWM, following a well-defined surgical and clinical approach, are able to support faster and complete healing of DFUs, that still represent a clinical challenge.

Methods and Materials

- Line 131: Refer to TASC II data for the diagnosis of peripheral vascular disease. (Norgren L, Hiatt WR, Dormandy JA, Nehler MR, Harris KA, Fowkes FG; TASC II Working Group. Inter-Society Consensus for the Management of Peripheral Arterial Disease (TASC II). J Vasc Surg. 2007 Jan;45 Suppl S:S5-67. doi: 10.1016/j.jvs.2006.12.037. PMID: 17223489)

Thank you for your suggestion. we have considered the referee's comments and will amend the text accordingly

- Lines 130-131. Why are patients with mild neuropathy excluded according to the Michigan rating? (<3)

Exclusion criteria included a Michigan Diabetic Neuropatthy Screening score (MDNS) <3.  Score <3 is not considered pathological and therefore we excluded those who did not have diabetic neuropathy

Neuropathy screening consisted of a 15-item questionnaire on and a brief clinical examination; patient those with a positive score >2 on clinical examination were considered to have neuropathy.

- Line 138. Define the clinical type of ulcer to choose IFWF or IDRT

Thank you for your suggestion.

10 patients with DFUs involving different tissues from epithelium to cartilage or bone structures (Wagner grade 3) were selected to receive acellular dermal matrix treatment. 5 with flat lesions without cavities were covered with IDRT and the other half of the lesions, which presented with cavities, were treated with IFWM a gel filling the cavity completely.

- Line 140-141. Justify why the atb times were so long (continued until the wound healed)

Thank you for your suggestion.

7/10 patients were suffering from osteomyelitis and the others were deep infections and the duration of antibiotic therapy should never be less than 30 days.Especially in cavity lesions treated with IFWM in which a material was injected and the cavity closed, it was essential to treat the patient with antibiotic therapy until healing

Results

Table 3. Baseline demographics and clinical characteristics. Should include p values to show homogeneity or heterogeneity between groups

the aim of the work is not a comparison between biomaterials applied in different types of lesions, therefore no statistical comparison was made. Moreover, the number of cases is 10 patients in total and any statistical analysis seemed useless given the small number of patients

Table 5. Secondary endpoints in IDRT and IFWM. Should include p values to show homogeneity or heterogeneity between groups

the aim of the work is not a comparison between biomaterials applied in different types of lesions, therefore no statistical comparison was made. Moreover, the number of cases is 10 patients in total and any statistical analysis seemed useless given the small number of patients

Discussion

It is probably too long.

Thank you for your suggestion. we have considered the referee's comments and will amend the text accordingly;

perhaps a little long but the study is complex and consists of a clinical part and a biochemical and histological analysis

Lines 5589-556. I recommend again referring to the meta-analysis of the role of MMOPS in ulcer healing.

Thank you for your suggestion.

we have considered the referee's comments and will amend the text accordingly
